# Physical and psychological health in intern paramedics commencing shift work: Protocol for an exploratory longitudinal study

Meagan E. Crowther[1]*, Sally A. Ferguson[1], Robert J. Adams[2], Katya Kovac[1], Jessica L. Paterson[3], Amy C. Reynolds[2]

1 Appleton Institute, CQUniversity, Wayville, South Australia, Australia, 2 Flinders Health and Medical Research Institute (Sleep Health), College of Medicine and Public Health, Flinders University, Bedford Park, South Australia, Australia, 3 Flinders Institute of Mental Health and Wellbeing, College of Education, Psychology and Social Work, Flinders University, Bedford Park, South Australia, Australia

* m.crowther@cqu.edu.au

## Abstract

**Data Availability Statement:** No datasets were generated or analysed during the current study. All relevant data from this study will be made available upon study completion.

### Background

Paramedics are routinely exposed to shift work. Existing research shows that shift work exposure is associated with adverse mental and physical health outcomes. However, the current understanding of the impact of commencing shift work in a paramedic role on health is limited. This can be addressed by recruiting new paramedics before they commence shift work, and conducting regular follow-ups of potential biological, psychological and social changes. The present study aimed to examine changes in biological, psychological and social factors relative to pre-shift work baseline in a cohort of paramedics commencing intern employment with an Australian ambulance service.

### Method and analysis

This observational, mixed-methods, longitudinal study aims to recruit 40 interns from one Australian ambulance service. Data collection will occur at baseline (standard day schedule for initial training), and subsequently at three months, six months, nine months and twelve months, to measure biological, psychological and social changes relative to baseline measurements. Changes in cardiometabolic markers (cholesterol, triglycerides, fasting glucose), microbiome (self-collected stool samples), sleep and physical activity (actigraphy) will be measured. Interns will also complete a battery of self-report questionnaires, sleep diaries and qualitative interviews to explore various psychological and social variables over time. Statistical analyses will be conducted using mixed effects regression, specifying a random effect of subject on the intercept, allowing participants to vary according to individual baseline levels, as well as tracking progress over time, appropriately accounting for serial correlation. Qualitative study components will be analysed via coding and thematic analysis procedures.

**Funding:** The study is funded by Central Queensland University internal Pilot Linkage Grant awarded to the research team with in-kind contributions from the South Australian Ambulance Service and Ambulance Employees Association SA, and an Australasian Sleep Association Rob Pierce Grant in Aid awarded to ACR. The lead author (MEC) is supported by an Australian Government Research Training Program scholarship. The financial funders had and will not have a role in study design, data collection and analysis, decision to publish, or preparation of the manuscript.

**Competing interests:** The authors have declared that no competing interests exist.

## Discussion

The present study protocol is a comprehensive outline of the observational study planned. The study will allow for greater knowledge of any changes in biological, psychological and social factors during a 12-month transition to shift work. The findings from the proposed study will have implications for the development of strategies to support early-career shift workers.

## Introduction

In order to meet the 24/7 demands of emergency health care, shift work is a key component of routine paramedic duties. For paramedics, this involves work that is scheduled throughout the 24h day. 'Shift work' encompasses a variety of work schedules including: fixed (morning, evening and/or night), variable, and/or rotating (a combination of two or more shift types) [1, 2]. Shift work, and particularly night work, means that sleep, wake, physical activity and meal timings need to occur at times of the day that are biologically inappropriate, which contributes to misalignment of various intrinsic rhythms [3, 4]. Such circadian misalignment is associated with a range of biological, psychological and social disruptions [2, 5].

Commonly reported chronic health concerns in shift workers with medium- to longer-term exposure with shift work include weight gain and obesity, type 2 diabetes, metabolic syndrome, and cardiovascular events [2, 6, 7]. Gastrointestinal distress [8, 9], poor mental health [10, 11], and somatic and musculoskeletal complaints [12, 13] are also more prevalent in shift working populations. In addition to chronic health concerns, shift workers commonly report insufficient and poor sleep [2, 14], as well as poorer health behaviours including higher levels of alcohol consumption (i.e. binge drinking) [15, 16], insufficient physical activity levels [17], and suboptimal nutritional intake [17, 18].

In addition to what is currently known about health outcomes in shift workers more broadly, evidence suggests that paramedics experience high levels of stress, depression, fatigue, burnout and increased risk of post-traumatic stress disorder [19–23]. A recent systematic review suggests that organisational support and individual coping mechanisms may protect against the development of these mental health risks in paramedics [24]. There is a need to understand psychosocial factors that may support good health during the transition to shift work. Collectively, there is a lack of longitudinal studies that investigate changes, positive or negative, in physical and psychological health outcomes in paramedics, particularly when commencing their career [25], which makes it challenging to understand the temporality of these experiences.

A better understanding of the effects of shift work on a paramedic's physical and psychological health is needed to determine associations with medium to long term health outcomes and if necessary, assist in early, preventive strategies to minimise adverse impacts of shift work on health and wellbeing.

To gain a complete picture of how shift work may be associated with changes in health over time, longitudinal follow-up across multiple time points, with a meaningful pre-exposure baseline is required. By examining changes from pre-shift work through the early months of a career, we can clarify important changes in biological, psychological and social factors for new paramedics.

### Objectives

Given the limited existing evidence, this study is exploratory in nature. However, an overarching study aim, and specific aims, have been identified. Research questions have been identified for qualitative measures.

**Overarching aim.**    The present study aims to characterise biological, psychological, and social changes during the paramedic intern year in a cohort of Australian paramedics.

**Hypothesis.**    It is hypothesised that the transition to shift work will be associated with changes in cardiometabolic risk, the gut microbiome, health behaviours and wellbeing, when compared to baseline.

**Specific aim 1.**    To determine whether there are clinically meaningful changes in cardio-metabolic risk factors during the first year of shift work.

**Specific aim 2.**    To characterise the relationship between onset of shift work and the gut microbiome.

**Specific aim 3.**    To explore changes in health behaviours (e.g. physical activity, alcohol, smoking and dietary intake) and sleep over the first year of shift work.

**Specific aim 4.**    To understand changes in psychosocial wellbeing during the first year of shift work.

**Research questions for qualitative measures.**

RQ1. What are the perceived workplace and personal strategies for coping with sleep inertia in paramedic interns, and do these change over the first year of shift work?

RQ2. What are the perceived workplace and personal strategies for coping with on-call aspects of paramedic work in interns, and how do these change over the first year of shift work?

RQ3. What do paramedic interns know about formal and informal workplace support for exposure to traumatic events experienced at work?

## Materials and methods

### Study design

This is an exploratory, longitudinal mixed-methods study. Data collection will be conducted for each intern at five time points: baseline (during intern training, and before commencing shift work), three months, six months, nine months and 12 months.

### Study population

Participants are newly employed intern paramedics (henceforth referred to as 'interns') from one Australian Ambulance service. Interns were chosen as the study population due to their regular requirement for shift work, the ability to capture a 'baseline' for each worker before commencing shift work, and the current, limited existing knowledge of the impact of emergency services work on biological, psychological and social factors during the transition to shift work [25]. Further, emergency personnel have a unique risk profile with their job requiring driving, delivering emergency health care and exposure trauma. Therefore, exploring the transition to emergency service work will provide crucial insights into a unique and vulnerable working population.

### Ethics approval

This project has been approved by the appropriate health department with reciprocal approval from the CQUniversity Human Research Ethics Committee, project number 0000022264.

### Study procedure

### Recruitment and consent

Due to the exploratory nature of this study, and in line with similar exploratory protocols [26], we have not calculated a specific sample size. We aim to recruit 40 interns to allow for

exploration of possible effects and necessary future sample sizes. Recruitment will be conducted in collaboration with South Australian Ambulance Service. All interns undertaking SAAS internship orientation during the study recruitment period will be invited to participate in the project by a senior researcher (ACR) within two weeks of commencing intern training. ACR will attend the training site during the intern orientation for each cohort, and provide a 30 minute presentation about the study. Following this presentation, interns will be able to ask any additional questions and sign up for the study if they choose. Participation is entirely voluntary, and intern paramedics are explicitly advised that participation is anonymous and will in no way impact their employment. Participants will only be recruited through the SAAS internship orientation, as outlined above.

Recruitment will continue with each new intake of interns until the target of $n = 40$ is reached. To minimise participant selection bias, all new interns commencing with South Australian Ambulance Service during the recruitment window will be invited to participate until the enrolment target is met.

**Inclusion criteria.**   Participants must be new recruits to South Australian Ambulance Service and have recently (within 2 years) graduated from a tertiary paramedic qualification which meets the Australian Health Practitioner Regulation Agency registration requirements for paramedicine.

**Exclusion criteria.**   Individuals who meet any of the following criteria will not be eligible to participate:

a. circumstances that interfere with the participant's ability to give informed consent (diminished understanding or comprehension, or a language other than English spoken and an interpreter unavailable),

b. Paid employment with another ambulance agency for >12 months at any time,

c. Unwilling or unable to provide quarterly participation.

The exclusion of participants based on their ability to give informed consent is necessary given the sensitive health information collected during the research project. Participants who have worked for another ambulance agency for >12 months are not considered "new" paramedics and are therefore a different population, and are excluded. Finally, exclusion based on participants being unwilling or unable to commit to quarterly participation, ensures inclusion of participants who are able to contribute data which addresses the aims of the intended study.

Experienced research personnel will obtain written, informed consent from eligible participants during the enrolment session. At the time of enrolment, each participant's email address and phone number are obtained for future contact.

## Status and timeline of the study

The study commenced rolling recruitment in November 2020. It is expected that recruitment will end in Quarter 2, 2022. Data collection will not be completed until 12 months after final recruitment (approximately Quarter 2, 2023). Some preliminary analysis of the first recruitments is planned for November 2022 for a doctoral thesis. However, final results and publications are not planned until all data collection is finalised.

## Data collection

Interns will be invited to complete biological (blood, stool samples), psychological and social (questionnaires and interviews) measures across five time points spanning 12 months. Table 1 provides an overview of which measures are collected at each time point. Contact will be

**Table 1. Time points for measures collected in the study.**

| Measure | Baseline (intern training) | 3 months | 6 months | 9 months | 12 months |
|---|---|---|---|---|---|
| **Sleep and activity measurement** | | | | | |
| Actigraphy (~7 days) | x | x | x | x | x |
| **Biological Measures** | | | | | |
| Pathology (fasting) | x | x | x | x | x |
| Stool Sample(s) | x | x | | | x |
| **Self-report Measures** | | | | | |
| Height and weight | x | | | | x |
| Questionnaires (see Table 2) | x | x | x | x | x |
| ASA24 food recall (via phone) [28] | x | x | x | x | x |
| Sleep diaries (7 Days) | x | x | x | x | x |
| NASA Task Load (7 Days) [29] | x | x | x | x | x |
| **Qualitative Measures** | | | | | |
| Occupational trauma exposure[+] | x | | | | |
| Sleep inertia awareness[+] | | x | | | |
| Napping and sleep inertia on night shift[+] | | x | x | x | x |

Note. ASA24; Automated Self-Administered 24-hour [+] Semi-structured interview script provided in S3–S5 Appendices.

Interns will be provided a $30 gift card at each data collection point (maximum $150) as acknowledgement of time spent participating in the study.

maintained with interns via REDCap [27] initiated emails, text messages and phone calls throughout the study to support completion.

T0: Baseline, prior to beginning shift work, completed during intern training

T1; 3 months after commencing the study (~6 weeks into shift work)

T2; 6 months after commencing the study

T3; 9 months after commencing the study

T4; 12 months after commencing the study

## Primary outcomes

**Cardiometabolic risk factors.** Fasting blood samples are collected at each data collection point to assess cholesterol and triglycerides, fasting blood glucose, cortisol, and c-reactive protein. At recruitment, interns are provided with a blood collection form with their unique identification code. Interns are informed that blood tests must be taken in the morning, while fasting, and cannot be collected during the morning immediately following a night shift. Interns can take the blood request to any clinic location associated with the pathology provider that is convenient for them. Interns are reminded via text message to undertake their blood test as required. Participation in blood tests at each follow-up is voluntary, and interns who elect not to participate will not be sent follow-up blood request forms.

All blood test results will be reviewed by the research team physician. If any blood test returns a clinically significant abnormal finding: 1) Interns will be contacted about their results by the research team physician, and 2) Results will be provided to the appropriate GP for follow up, providing participant consent has been given.

Ethical approval stipulated that interns be allowed to participate in the study without completion of pathology, so interns could continue in follow ups if they chose, without continuing pathology measures. Therefore, there may not be pathology data for all participants.

**Stool samples.**   Stool samples will be collected to assess the structure and function of the gastrointestinal microbiome, as this has recently been proposed as a mechanism linking shift work with metabolic health outcomes later in life [4, 30]. Interns will be asked to provide stool samples for analysis at baseline, the first follow-up and the 12-month (final) follow-up. Included with their stool sample collection kit are instructions, the stool collection tube (Norgen Biotek Corp, ON, Canada) and a flushable sample collection kit for convenient sample collection (HyStool, Aberlady, Scotland). A maximum of three samples will be requested per data collection point. Interns will be instructed at baseline and one month to collect their sample at their convenience. At 12 months, interns will be asked to provide three samples: one after a series of days off shift, one after a series of day shifts, and one after a series of night shifts to allow for assessment of temporal changes to the microbiome related to different shift schedules. Samples will be collected using Stool Nucleic Acid collection and preservation tubes (Norgen Biotek Corp, ON, Canada), allowing them to be stored at ambient temperature and removing the necessity for cold-chain transportation. A brief (~30 second) checklist will be completed for each sample to allow for interpretation of findings (see S1 Appendix). Interns will be reminded via text message to undertake and return stool samples as required.

### Secondary outcomes

**Objective sleep and physical activity.**   Activity monitors (GENEActiv, Actvinsights, UK) will be used at each time point to objectively measure sleep and physical activity. Interns will be sent a pre-programmed activity monitor at each data collection point. When the activity monitor has been posted by a research team member, a text message will be sent to the intern to inform them that it should arrive in a few days and to put the activity monitor on as soon as possible, and wear it for a minimum of seven days, before returning it in the provided pre-addressed and tracked postage satchel. Activity monitors will be worn in order to measure overall sleep and physical activity, and measure sleep in relation to certain shifts.

**Self-report measures.**   The self-report measures described below are all collected via a REDCap [27, 31] survey. An overview of the items collected in the electronic survey is detailed in Table 2. Following enrolment, interns are emailed an invitation for the baseline survey. Interns can pause and restart the survey at any time by selecting "Save & Return", which will email an updated survey link to the designated email account. If the survey is not initiated within seven days, a reminder email is sent via REDCap [27, 31]. Reminders are sent to interns until the survey is completed. Interns will also be reminded to complete the survey via text message or during quarterly qualitative interviews, as necessary.

**Shift system information.**   At baseline, to reduce unnecessary participant burden, a modified Standard Shift Work Index (SSI) [32] will be used. In the modified version, interns are asked to report their standard start and finish time of each training day (rather than individual shift type) as they are not yet working shift work. Shift system details are obtained at each time point after commencing shift work (T1 to T4) using the original SSI [32]. Interns are asked to list the normal start and end times of each of their shifts (e.g. morning, afternoon, night shifts). The SSI also asks interns to describe their preference of shifts and reasons for working shift work.

**Sleep.**   Self-reported sleep will be investigated using the SSI [32] and the Pittsburgh Sleep Quality Index (PSQI) [40]. Further, to assess daily sleep time and quality, interns are asked to complete sleep diaries for seven consecutive days. Unlike the survey, no reminders are sent for sleep diaries to avoid over burdening interns with contact at each of the quarterly follow-ups. The use of multiple measures of sleep allows for assessment of self-reported habitual sleep (PSQI), and changes relative to shift schedules. The SSI asks interns to recall their habitual

**Table 2. Overview of characterises of self-administered measures.**

| Instrument | Captures | Measured at: | Time to self-administer | Planned Scoring | Interpretation (↑, *higher score*) |
|---|---|---|---|---|---|
| **Standard Shiftwork Index (SSI)** [32] | Work context and shift system details General job satisfaction [33] Sleep Chronic Fatigue Physical health Psychological health (General health questionnaire) [34] Cognitive-somatic Anxiety [35] Social and domestic situation Coping Circadian rhythm type (Composite morningness questionnaire) [36] Circadian type (Circadian type inventory) Eysenck Personality Inventory [37] | T0, T1, T2, T3, T4 | 30 mins | Job satisfaction: 0–7 Sleep disturbance: 0–5 Chronic Fatigue: 5–50 Digestive health: 8–32 Cardiovascular health: 8–32 Psychological wellbeing: 0–36 Cognitive anxiety: 7–35 Somatic anxiety: 7–35 Social and domestic satisfaction: 21–105 Morningness Scale: 13–55 Circadian languidity: 10–50 Circadian flexibility: 8–40 Neuroticism: 6–24 Extraversion: 6–24 | ↑ = more satisfied ↑ = sleep disturbance ↑ = more fatigue ↑ = digestive problems ↑ = cardiovascular problems ↑ = poorer psychological health ↑ = greater anxiety ↑ = greater anxiety ↑ = more satisfaction ≤ 22 evening type 23–43 intermediate type ≥ 44 morning type ↑ = more circadian languidity ↑ = higher circadian flexibility ↓ = higher neuroticism ↓ = high extraversion |
| **Shift work disorder screening questionnaire** [38] | 4-item screening tool to aid in diagnosis of shift work disorder. | T0, T1, T2, T3, T4 | ~5 mins | In accordance with criteria specified by the developing authors [38] | ↑ = likelihood of SWD |
| **Short Form General Health survey (SF-20)** [39] | 20-item measures of physical functioning and wellbeing. | T0, T1, T2, T3, T4 | ~5 mins | Physical Functioning: 0–100 Role Functioning: 0–100 Social Functioning: 0–100 Mental Health: 0–100 Health Perceptions: 0–100 Pain: 0–100 | ↑ = better health |
| **Pittsburgh Sleep Quality Index (PSQI)** [40] | Assess sleep quality, sleep latency, sleep duration, habitual sleep efficiency, sleep disturbances, use of sleeping medication and daytime dysfunction for previous 30 days. | T0, T1, T2, T3, T4 | 5–10 mins | Global Sleep Score: 0–21 Subjective Sleep Quality: 0–3 Sleep latency: 0–3 Sleep duration: 0–3 Habitual sleep efficacy: 0–3 Sleep disturbances: 0–3 Sleep medication: 0–3 Daytime dysfunction: 0–3 | ↑ = poorer sleep quality Global sleep score > 5 = clinically poor sleep |
| **Occupational Fatigue and Exhaustion Recovery Scale (OFER15)** [41] | 15-item scale measuring chronic work-related fatigue, end-of-shift states and recovery between shifts. | T0, T1, T2, T3, T4 | 5 mins | Chronic faitue:0–100 Acute fatigue: 0–100 Inter-shift recovery: 0–100 | ↑ = more chronic fatigue ↑ = more acute fatigue ↑ = more recovery between shifts |
| **International Physical Activity Questionnaire (IPAQ)** [42] | 27-item measure of physical activity for past seven days. Job-related physical activity, transportation, housework/house maintenance/family caring, sport/leisure time activity and sedentary time. | T0, T1, T2, T3, T4 | 15–20 mins | Walking: 3.3 METs Moderate activity: 4 METs Vigorous activity: 8 METs | ↑ = more physical activity or more intensive physical activity May be categorised in to High, Moderate Low Physical Activity levels |
| **Post-traumatic stress disorder checklist for DMS-5 (PCL-5)** [43] | 20-item measure that assess DMS-5 symptoms of post-traumatic stress disorder. | T0, T1, T2, T3, T4 | 5–10 mins | Symptom severity:0–80 | ↑ = higher symptom severity |
| **Newest Vital Sign (NVS)** [44] | 6-item measure of health literacy, using nutritional label. | T0, T1, T2, T3, T4 | 3 mins | Health Literacy: 0–6 | ↑ = better health literacy |
| **Shift worker risk perception questions** | 5-item measure intended to measure health risk perception in shift workers | T0, T1, T2, T3, T4 | 3 mins | Health risk perception: 5–25 | ↑ = higher levels of perceived health risk |

(*Continued*)

**Table 2.** (Continued)

| Instrument | Captures | Measured at: | Time to self-administer | Planned Scoring | Interpretation (↑, *higher score*) |
|---|---|---|---|---|---|
| **Sleep diaries– 7 days** | 24-hour period. Time to bed, time asleep, time awake, time out of bed, total sleep time, waking during sleep period, length of waking, sleepiness prior to sleep, sleepiness upon waking, self-report sleep quality, naps (Y/N), length of naps (if relevant). | T0, T1, T2, T3, T4 | ~5 mins | Total sleep time<br>Time in bed<br>Sleep latency<br>Sleepiness prior to bed<br>Sleepiness on waking<br>Sleep quality<br>Naps | NA |
| **NASA Task load (NASA-TLX) [29]– 7 days (when on shift)** | Shift start/finish time.<br>6-item workload assessment measuring mental demand, physical demand, temporal demand, performance, effort, frustration. | T0, T1, T2, T3, T4 | ~3 mins | Workload: 0–100 | ↑ = higher workload<br>Low: 0–9<br>Medium: 10–29<br>Somewhat high: 30–49<br>High: 50–79<br>Very high: 80–100 |

**Note.** DSMV = Diagnostic and Statistical Manual of Mental Disorders, METs = Metabolic equivalent of task.

sleep on specific shifts, which allows exploration of how different shifts may be impacting interns' sleep, which is not covered by the PSQI.

**Physical activity.** Self-report physical activity will be measured by the International Physical Activity Questionnaire (IPAQ) [42]. The IPAQ assesses work-related physical activity, as well as recreational activity and sitting time, which is important as the interns may experience changes in both work-related and leisure-time physical activity throughout the 12-month period.

**Dietary intake.** Dietary intake data will be collected using the Automated Self-Administered 24-hour Dietary Assessment Tool–Australia 2016 (ASA24), developed by the National Cancer Institute [28]. The ASA24 is completed via phone, with a member of the research team asking interns to recall what they have consumed, where the food/drink was purchased, and the time of day of consumption. The ASA24 allows for calculation of both total 24-hour caloric consumption and nutritional values such as: sodium, protein, fat, sugar, carbohydrates, fibre and iron. The ASA24 also records timing of food consumption, which is relevant for interns who may be eating outside the biological norm [18].

**Perceived health and wellbeing.** Changes in perceived health and wellbeing will be assessed using various scales within the self-report data. The SF-20 [39] and SSI [32] will report measures of perceived wellbeing and physical health. The shift work disorder screening questionnaire [38] will identify interns who may be experiencing shift work disorder symptoms. Further, interns will complete the post-traumatic stress disorder checklist (PCL-5) [43] which examines possible PTSD symptoms given increased risk identified in the existing paramedic literature [19].

**Occupational load.** Following initiation of each data collection survey, the NASA Task Load Index (NASA-TLX) [29] will be sent together with self-reported sleep diaries to interns via REDCap [27, 31] for seven consecutive days. No reminders are sent for the NASA-TLX. Before completing the NASA-TLX, each intern is asked if they worked yesterday (Yes or No), and if they did not work they will not be asked to fill out a NASA-TLX for that day. When interns indicate that they have worked a shift the previous day, they are prompted to enter their shift start and finish times which allows for calculation of which shift they worked (e.g. night shift). Interns are then prompted to "Please answer these questions in regard to your overall work shift yesterday", with standard NASA-TLX questions to follow.

**Occupational fatigue and recovery.** Fatigue associated with work and recovery in between shifts will be assessed using the OFER15 [41] at all quarterly data collection points.

**Health literacy.** The Newest Vital Sign [44] will be used to quantify the baseline health literacy of new intern paramedics and whether this changes over the course of 12-months work in healthcare.

**Shift work risk perception.** This 5-item measure was developed by the present study author (MEC), and is based on both previous evidence of health risk perception in shift workers [45] and the cardiovascular risk perception scale [46]. The scale aims to examine whether shift workers perceive shift work as a risk to their health (S2 Appendix), and whether this changes with onset of shift work.

**Qualitative interview.** At each data collection point, interns are contacted via text message to organise a convenient time for an interview. At the arranged time, interns are called by a research team member. Following completion of the ASA24, an audio recording device is turned on and interns are asked the appropriate interview questions for the time point (Table 2)—full semi-structured qualitative scripts are available in S3 and S4 Appendices. Baseline qualitative interviews will aim to investigate how intern paramedics perceive exposure to occupational trauma, and ask them to identify possible support strategies (S3 Appendix). The first qualitative interview following commencement of shift work (Time 1) will ask interns to recall their experience of sleep inertia and whether they have had any education regarding sleep inertia (S4 Appendix). Finally, all interviews from commencement of shift work (Time 1) will ask interns about their experience of napping, sleep inertia and performance on night shift which existing literature suggests may influence recovery and sleep in paramedics [47] (S5 Appendix). Further, interns will be asked to describe any strategies they use to combat sleep inertia to further understand countermeasures for the decline in performance upon waking associated with sleep inertia [48].

## Statistical analysis

Analysis will be conducted to investigate any changes in i) cardiometabolic risk factors, ii) human gut microbiome, iii) health behaviours (e.g. physical activity, alcohol, smoking, dietary intake) and sleep, and iv) wellbeing during first year of shift work. Missing data will be addressed using multiple imputation, as appropriate. Descriptive statistics will be included to provide information regarding participant characteristics. To examine any changes over time, statistical modelling will be conducted using mixed effects regression, specifying a random effect of subject on the intercept, allowing interns to vary according to individual baseline levels, as well as tracking progress over time, appropriately accounting for serial correlation. Effect sizes and 95% confidence intervals will be considered during analyses. Key descriptive statistics (e.g. demographics, sleep and health at baseline) will be compared between participants lost to follow-up and those who continued within the study, using t-test or non-parametric equivalent if necessary, to determine any factors which differ between the two groups.

Qualitative interviews will be explored with thematic analysis to investigate i) perceived workplace and personal strategies for coping with sleep inertia, ii) perceived workplace and personal strategies for coping with on-call work, and iii) knowledge of formal and informal workplace supports for exposure to traumatic events experienced in their job role. Qualitative interview data will be managed using NVivo software [49] and thematically coded [50] and analysed using a general inductive approach since there are no prior hypotheses or assumptions.

## Data management plan

Self-report data will be collected electronically via REDCap. All interview data, identifiable REDCap data and/or pathology results will be stored in a secured CQUniversity research

drive. Only the immediate research team members will have access to this research drive. Each participant will be allocated a participant ID in order of recruitment, which will be used to de-identify all data. Paper copies of consent forms and any other paper results will be stored in a secure locked cabinet at Flinders University.

## Dissemination

Final data analysis will begin after completing data collection. Preliminary analyses will be conducted on a partial sample for the purpose of PhD dissertations. Study results will be disseminated through peer-reviewed journal publications and national and international conference presentations.

## Discussion

### Summary

Shift work is associated with negative health outcomes. Our understanding of possible underlying mechanisms for, or protective factors against, these medium- to long-term health outcomes, and the possible onset of these mechanisms during the transition to shift work are not well understood. This means we have no clear understanding of the sequalae of health changes, positive or negative, which impacts our ability to provide preventive advice and develop strategies for early career shift workers. The proposed study facilitates frequent, intensive, observational data collection during the 12-month internship program in the South Australian Ambulance Services internship program. Crucially, these findings will provide new knowledge of possible biological, psychological, and social changes during the transition to shift work, with likely applications beyond paramedicine to other shift working professions.

### Strengths of the planned study

The proposed study is the first known study to utilise quarterly data collection together with a pre-shift work baseline to explore physical and psychological health changes across the first 12 months of paramedic employment. The aim of our study is to comprehensively examine biopsychosocial changes, within individuals, during the transition into shift work. These findings study will strengthen our understanding of the effect of commencing shift work on biological, psychological, and social factors early in shift work employment.

### Limitations

The present study benefits from collecting data in the field, however, with this comes the limitation that we cannot attribute all changes to what we are measuring within the study. It is possible that changes may occur longitudinally that are not related to the transition to shift work. A further limitation is that the present study is conducted at a single site with interns recruited from one Australian ambulance service, therefore the outcomes may be less applicable across different sites or areas. However, given this is an exploratory study, it may be possible to extend across various sites and locations in future. The present study will be susceptible to selection bias. In order to best mitigate this, all eligible interns during the recruitment period will be offered the opportunity to participate, and we will transparently report any baseline differences in those who are lost to follow-up, compared to those who remain in the study. We will not be able to compare differences in those who did not consent to the study, as we do not have data on the interns not included in the present study. This risk of bias will be acknowledged in any reporting of findings from the study to provide context.

## Conclusion

The proposed study will allow for better understanding of biopsychosocial changes during the transition to shift work. By conducting this exploratory, observational study we may better understand if any changes, positive or negative, occur during early career shift work.

## Supporting information

**S1 Appendix. Microbiome checklist.**
(DOCX)

**S2 Appendix. Shift work risk perception scale.**
(DOCX)

**S3 Appendix. Occupational trauma exposure semi-structed qualitative script.**
(DOCX)

**S4 Appendix. Sleep inertia education experience semi-structed qualitative script.**
(DOCX)

**S5 Appendix. Napping and sleep inertia during night shifts semi-structed qualitative script.**
(DOCX)

## Acknowledgments

The authors wish to acknowledge members of the research team who contributed to development of research questions included in this study, including Chris Howie (SA Ambulance Service), K Townsend & R Loudoun (Griffith University).

## Author Contributions

**Conceptualization:** Meagan E. Crowther, Sally A. Ferguson, Robert J. Adams, Katya Kovac, Jessica L. Paterson, Amy C. Reynolds.

**Funding acquisition:** Sally A. Ferguson, Robert J. Adams, Jessica L. Paterson, Amy C. Reynolds.

**Investigation:** Meagan E. Crowther, Sally A. Ferguson, Amy C. Reynolds.

**Methodology:** Meagan E. Crowther, Sally A. Ferguson, Robert J. Adams, Katya Kovac, Amy C. Reynolds.

**Project administration:** Meagan E. Crowther, Robert J. Adams, Katya Kovac, Amy C. Reynolds.

**Resources:** Meagan E. Crowther, Sally A. Ferguson, Robert J. Adams, Amy C. Reynolds.

**Software:** Meagan E. Crowther, Amy C. Reynolds.

**Supervision:** Sally A. Ferguson, Jessica L. Paterson, Amy C. Reynolds.

**Writing – original draft:** Meagan E. Crowther, Sally A. Ferguson, Robert J. Adams, Jessica L. Paterson, Amy C. Reynolds.

**Writing – review & editing:** Meagan E. Crowther, Sally A. Ferguson, Robert J. Adams, Katya Kovac, Jessica L. Paterson, Amy C. Reynolds.

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
