## [Decision Letter · Decision Letter 0]

22 Jul 2022

PONE-D-22-14838Physical and psychological health in intern paramedics commencing shift work: Protocol for an exploratory longitudinal studyPLOS ONE

Dear Dr. Crowther,

Thank you for submitting your manuscript to PLOS ONE. After careful consideration, we feel that it has merit but does not fully meet PLOS ONE’s publication criteria as it currently stands. Therefore, we invite you to submit a revised version of the manuscript that addresses the points raised during the review process.

There are several suggestions made that once addressed, the manuscript would like meet the criteria for publication. I anticipate that the authors may not agree with all of these, and considered rebuttal would be acceptable. More detailed description of recruitment and sampling is necessary though. 

We look forward to receiving your revised manuscript.

Kind regards,

Dylan A Mordaunt, MD, MPH, FRACP

Academic Editor

PLOS ONE

Journal Requirements:

"ACR has received research grant funding from a variety of sources over the course of research relating to this manuscript, including the Sleep Health Foundation (including an unrestricted grant from Merck, Sharp & Dohme Australia and Carers Australia via the Sleep Health Foundation), Vanda Pharmaceuticals, Compumedics, the Australasian Sleep Association, the Hospital Research Foundation, Flinders Foundation, Safework SA, Sydney Trains, Arthritis Australia, Bundaberg Regional Council, Queensland Fire and Emergency Services and the Freemasons Foundation (SA). ACR has received personal fees for work related to her research from Sealy Australia and Teva Pharmaceuticals. Professor Robert Adams (RJA) reports grants from the Sleep Health Foundation, Philips Respironics, The National Health and Medical Research Council, ResMed Foundation, The Hospital Research Foundation, Flinders Foundation, Sydney Trains and the Research Network for Undersea Decision Superiority during the past 3 years.

No other authors have competing interests to declare."

Additional Editor Comments:

Thank you for your submission.

1) We've received a single review that points to primarily issues either with method description, or with discussion of limitations of the method, mostly related to sampling.

2) It would be helpful for the authors to address with some detail the source population, approach to sampling (e.g. snowball, convenience etc) in a PICOT manner.

3) There are notable exclusions, and although explanations of exclusions aren't necessarily standard elements of reporting checklists for survey studies, a description of the rationale behind these criteria would fit with quality rating tools for other article types and I think would help improve the quality of the paper.

4) The reviewer also raises the issue of a framing effect or bias. The disclosures are noted and as stated the primary funder is an independent entity with a public health remit. I think the authors could consider their framing with regards to the broad readership of PLoS One, and aim to present a more neutral setup prior to establishing the case that they conclude from the data. This is not merely a semantic suggestion, since rating tools for assessed quality of prospective cohort studies include risk of bias assessments (e.g. https://methods.cochrane.org/risk-bias-2).

With specific regards to the criteria for publication:

1. The study appears to present the results of an original protocol (an accepted article type at PLoS One).

2. Results are not reported, as it's a protocol.

3. Experiments, statistics, and other analyses are performed to a reasonable technical standard but warrant attention to the areas described above and by the reviewer.

4. Conclusions are presented in an appropriate fashion and are supported by the data.

5. The article is presented in an intelligible fashion and is written in standard English.

6. The research meets all applicable standards for the ethics of experimentation and research integrity.

7. The article adheres to appropriate reporting guidelines and community standards for data availability, with the exception of the comments made above. I would suggest the authors consider the STROBE (https://www.equator-network.org/reporting-guidelines/strobe/) and SPIRIT checklists (https://www.equator-network.org/reporting-guidelines/spirit-2013-statement-defining-standard-protocol-items-for-clinical-trials/), acknowledging that this study does not meet criteria for these, but could help improve the study protocol quality. As stated above, it might also be worth considering assessed bias in your design, through a risk of bias tool such as cited.

NB: I have noted a disclosure to the staff editor that some of the authors and I have shared affiliations. I am an employee of Southern Adelaide Local Health Network, a clinician (medical practitioner) and am affiliated with Flinders University, but have noted that I have no past, current or planned collaborations with any of the authors. This statement is included for transparency.

Reviewers' comments:

Reviewer's Responses to Questions

**Comments to the Author**

1. Does the manuscript provide a valid rationale for the proposed study, with clearly identified and justified research questions?

Reviewer #1: Yes

2. Is the protocol technically sound and planned in a manner that will lead to a meaningful outcome and allow testing the stated hypotheses?

Reviewer #1: Partly

3. Is the methodology feasible and described in sufficient detail to allow the work to be replicable?

Reviewer #1: Yes

4. Have the authors described where all data underlying the findings will be made available when the study is complete?

Reviewer #1: Yes

5. Is the manuscript presented in an intelligible fashion and written in standard English?

Reviewer #1: Yes

6. Review Comments to the Author

You may also provide optional suggestions and comments to authors that they might find helpful in planning their study.

Reviewer #1: Thanks for the opportunity to review and apologies for the delay.

Congratulations on your protocol on this essential and currently increasingly evidenced area.

The aims and objectives are clearly defined. A hypothesis should be provided.

The risk of selection bias is always present in these types of studies, particularly when participation is voluntary. I do not see a way around this, but worth considering in the analysis/discussion.

The columns in table 1 should be changed to the actual times, not just T0,1,2,3,4

The caption in table 1 is to be moved to methods.

Table 2 lists the surveys to be used. This seems excessive and may result in high drop-out rates due to survey fatigue. Consideration about consolidating this list will likely lead to a more complete, although truncated data set.

The statistical analysis plan lacks details.

Reading the paper, I get the sense that the authors want to demonstrate the extent of the negative impact of shift work. Rather they could take a more unbiased approach and aim to determine how commencing shift work impacts health in any direction.

7. PLOS authors have the option to publish the peer review history of their article (what does this mean?). If published, this will include your full peer review and any attached files.

Reviewer #1: No

---

## [Author Response · Author response to Decision Letter 0]

1 Aug 2022

Please see response to reviewer document for detailed changes in respone to feedback from reviewer.

---

## [Editor Report · Decision Letter 1]

15 Sep 2022

Physical and psychological health in intern paramedics commencing shift work: Protocol for an exploratory longitudinal study

PONE-D-22-14838R1

Dear Dr. Crowther,

We’re pleased to inform you that your manuscript has been judged scientifically suitable for publication and will be formally accepted for publication once it meets all outstanding technical requirements.

Kind regards,

Jianhong Zhou

Staff Editor

PLOS ONE

Additional Editor Comments (optional):

Thank you for your resubmission. This now meets the criteria for publication.
---

## [Editor Report · Acceptance letter]

21 Sep 2022

PONE-D-22-14838R1 

Physical and psychological health in intern paramedics commencing shift work: Protocol for an exploratory longitudinal study 

Dear Dr. Crowther:

I'm pleased to inform you that your manuscript has been deemed suitable for publication in PLOS ONE. Congratulations! Your manuscript is now with our production department. 

Kind regards, 

on behalf of

Jianhong Zhou 

Staff Editor

PLOS ONE